# Discontinuous transcription of ribosomal DNA in human cells

**Evgeny Smirnov**[1]*, **Peter Trosan**[2], **Joao Victor Cabral**[2], **Pavel Studeny**[3], **Sami Kereïche**[1], **Katerina Jirsova**[2], **Dušan Cmarko**[1]

1 Laboratory of Cell Biology, Institute of Biology and Medical Genetics, First Faculty of Medicine, Charles University and General University Hospital in Prague, Prague, Czech Republic, 2 Laboratory of the Biology and Pathology of the Eye, Institute of Biology and Medical Genetics, First Faculty of Medicine, Charles University and General University Hospital in Prague, Prague, Czech Republic, 3 Ophthalmology Department of 3rd Faculty of Medicine, Charles University and University Hospital Kralovske Vinohrady, Prague, Czech Republic

* esmir@lf1.cuni.cz

**Data Availability Statement:** URL to access data: https://osf.io/2v8am/?view_only=1d925a4e3ce845b599cdc7908edd0aa1.

**Funding:** The work was supported by research project BBMRI_CZ LM2018125, European

## Abstract

Numerous studies show that various genes in all kinds of organisms are transcribed discontinuously, i.e. in short bursts or pulses with periods of inactivity between them. But it remains unclear whether ribosomal DNA (rDNA), represented by multiple copies in every cell, is also expressed in such manner. In this work, we synchronized the pol I activity in the populations of tumour derived as well as normal human cells by cold block and release. Our experiments with 5-fluorouridine (FU) and BrUTP confirmed that the nucleolar transcription can be efficiently and reversibly arrested at +4˚C. Then using special software for analysis of the microscopic images, we measured the intensity of transcription signal (incorporated FU) in the nucleoli at different time points after the release. We found that the ribosomal genes in the human cells are transcribed discontinuously with periods ranging from 45 min to 75 min. Our data indicate that the dynamics of rDNA transcription follows the undulating pattern, in which the bursts are alternated by periods of rare transcription events.

## Introduction

Numerous studies show that genes in all kinds of organisms, from prokaryotes to mammals, can be transcribed in short bursts or pulses alternated by periods of silence (reviewed in Smirnov et al. [1]) The probability of such mode of expression was suggested long ago;[2] now it seems that the discontinuous transcription is a common feature of the gene expression, at least in mammalian cells.[3–12] The periodical switches of the promoter between the active and "refractory" states may be crucial in the efficient regulation of the gene expression.[13–17] General considerations suggest even more significant role of the phenomenon in the dynamic organization of the cell, since the pulsing mode of one process is likely to be a cause and a consequence of pulsing in other processes. Thus, RNA processing, which is closely linked to the RNA synthesis, seems to be discontinuous.[9] A spontaneous heterogeneity of gene expression occasioned by transcriptional fluctuations may influence cell behaviour in changing environmental conditions and in the course of differentiation.[18]

Regional Development Fund, project EF16_013/0001674, by the Grant Agency of Czech Republic (19-21715S) and by Charles University (Progres Q25 and Q28). SK acknowledges the financial support from the Czech Science Foundation Grant No. 1825144Y. The funders had no role in study design, data collection and analysis, decision to publish, or preparation of the manuscript.

**Competing interests:** The authors have declared that no competing interests exist.

The discontinuous character of transcription has been detected by various methods (reviewed in Smirnov et al. [1]) The number of transcripts produced in a certain (sufficiently short) period of time may be determined with high precision by single molecule RNA fluorescence in situ hybridisation (smFISH).[19–21] The results of such quantification alone provide indirect, but valuable information for modelling the expression kinetics in a cell population or tissue, when the studied gene is supposed to be transcriptionally active in all the cells. Methods based on the allele-sensitive single-cell RNA sequencing also allow to reveal and characterize the transcription bursting.[22] To monitor gene expression in real time, cells are transfected with constructs providing a fluorescent signal that corresponds to the expression of a particular gene. In a gene trap strategy, a luciferase gene is inserted under the control of endogenous regulatory sequences. Since both the luciferase protein and its mRNA are short-lived, the method allows to calculate the key parameters of the transcriptional kinetics. Probably the most popular *in vivo* method is based on the use of bacteriophages derived fluorescent coat proteins, such as MS2 or PP7, fused with GFP, which allows to visualize a bunch of the nascent RNA molecules accumulated around one gene.[4, 23, 24]

So far, the pulse-like transcription is well documented only in the genes transcribed by RNA polymerase II. It is not clear yet whether ribosomal DNA (rDNA) is also expressed discontinuously. In human cells, the clusters of multiple rDNA repeats, known as Nucleolus Organizer Regions (NORs), are situated on the short arms of the acrocentric chromosomes. Each repeat includes a gene coding for 18S, 5.8S and 28S RNAs of the ribosomal particles and an intergenic spacer.[25–30] In the interphase nucleus the rDNA provides the basis for the formation of nucleoli. The transcription by pol I and the first steps of rRNA processing take place in the special nucleolar units (FC/DFC) composed of fibrillar centers (FC) and dense fibrillar components (DFC).[31–42] The units correspond in light microscopy to the "beads" forming nucleolar necklaces,[43–46] and each unit is believed to accommodate a single transcriptionally active gene.[33, 39, 47, 48] The intensity of the rDNA transcription is usually very high throughout the entire interphase, especially at the S and G2 phases.[49] Now most of the methods used for the detection of the transcription fluctuation are hardly applicable to the ribosomal genes, since one cell usually contains hundreds of such genes. An alternative method was designed for direct measurements of rDNA transcription in the live cells by using the label-free confocal Raman microspectrometry.[50] This work revealed an undulatory character of the ribosomal RNA production in the whole nucleoli. In our earlier study on tumour-derived cells expressing a GFP-RPA43 (a subunit of pol I) fusion protein, we have observed specific fluctuations of the fluorescence signal in the individual FC/DFC units.[51] We also found high correlation of pol I and incorporated FU signals within the units. These data suggested that the ribosomal genes are transcribed in a pulse-like manner.

In the present work we used a different approach to the study of the discontinuous transcription of ribosomal genes in human cells. In our experiments with 5-fluorouridine (FU) and BrUTP, we found that the nucleolar transcription can be efficiently arrested at +4°C and quickly restored at normal conditions. Based on this finding, we synchronized the pol I activity in the cell population by cold block and release. Then using specially designed software we measured the intensity of transcription signal (incorporated FU) in the nucleoli and individual FC/DFC units at different periods after the release. This enabled us to detect transcription fluctuations of ribosomal genes in tumour derived as well as normal human cells and to reveal special properties of this fluctuation.

## Methods

### Ethics

The study followed the standards of the Ethics Committees of the General Teaching Hospital and the First Faculty of Medicine of Charles University, Prague, Czech Republic (Ethics

Committee of General Univeristy Hospital, Prague approval no. 1570/11 S-IV (held on October 13, 2011, and updated January 18, 2018. The name of project: Pathogenesis of hereditary, degenerative and systemic diseases with manifestations in the eye, transplantology. Study of healthy and control tissue), and adhered to the principles set out in the Helsinki Declaration. We obtained human cadaver corneoscleral rims from 10 donors, which were surplus from surgery and stored in Eusol-C (Alchimia, Padova, Italy), from the Department of Ophthalmology, General University Hospital in Prague, Czech Republic, for the study. On the use of the corneoscleral rims, based on Czech legislation on specific health services (Law Act No. 372/2011 Coll.), informed consent is not required if the presented data are anonymous in the form."

## Cell cultures

Human limbal epithelial cells (LECs) were obtained from XY cadaver corneoscleral rims after cornea grafting at University Hospital Kralovske Vinohrady, Prague, Czech Republic. The mean donor age ± standard deviation (SD) was 63.5 ± 6.5 years. Tissue was stored in Eusol-C (Alchimia, srl., Ponte San Nicolò, Italy) preservation medium at +4˚C. The mean storage time ± SD (from tissue collection until explantation) was 7.2 ± 3.6 days. The corneoscleral rims were prepared as described before.[52, 53] Shortly, corneoscleral rims were cut into 12 pieces and placed in a 24-well plate (TPP Techno Plastic Products AG, Trasadingen, Switzerland) on Thermanox plastic coverslips (Nunc, Thermo Fisher Scientific, Rochester, NY, USA). Explants were cultured in 1 ml of complete medium [1:1 DMEM/F12, 10% FBS, 1% AA, 10 ng/ml recombinant EGF, 0.5% insulin-transferrin-selenium (Thermo Fisher Scientific), 5 µg/ml hydrocortisone, 10 µg/ml adenine hydrochloride and 10 ng/ml cholera toxin (Sigma-Aldrich, Darmstadt, Germany)]. The culture media were changed every 2–3 days until the cells were 90–100% confluent (after 2–4 weeks).

HeLa cells were cultivated at 37˚C in Dulbecco modified Eagle's medium (DMEM, Sigma) containing 10% fetal calf serum, 1% glutamine, 0.1% gentamicin, and 0.85g/l NaHCO$_3$ in standard incubators. For the transcription synchronization, the cells were incubated in cold medium (+4˚C) for 1 h, then transferred to the normal conditions and fixed at different time points from 15 to 150 min with the interval of 15 min.

## Plasmids and transfection

The GFP-RPA43 and GFP-Fibrillarin vectors were received from Laboratory of Receptor Biology and Gene Expression Bethesda, MD.[54] The constructs were transfected into HeLa cells using Fugene (Qiagen).

## Labeling of the transcription sites

For visualization of the transcription sites, sub-confluent cells were incubated for 5 min prior to fixation with 5-fluorouridine (FU) (Sigma). The cells were fixed in pure methanol at -20˚C for 30 min and processed for FU immunocytochemistry. BrUTP (Sigma) was introduced into cells by the scratch procedure.[55, 56] Here we followed the same procedure as for the labelling of replication in the cited works. Briefly, the cells were grown on the coverslips; a drop of medium containing 20µg/ml BrUTP was applied upon each coverslip; then the latter was scratched by the tip of a syringe needle and incubated for 5 min at 37˚C. Thus permeabilized, the cells were incubated for 10 min in the usual medium and then fixed and processed as after the incorporation of FU.

Incorporated FU and BrUTP signal was visualized using a mouse monoclonal anti-BrdU antibody (Sigma) and secondary goat Cy3-conjugated anti-mouse antibody (Abcam).

## Light microscopy

Confocal images were acquired by means of SP5 (Leica) confocal laser scanning microscope equipped with a 63×/1.4NA oil immersion objective. For *in vivo* cell imaging we used a spinning disk confocal system based on Olympus IX81 microscope equipped with Olympus UPlanSApo 100×/1.4NA oil immersion objective, CSU-X spinning disk module (Yokogawa) and Ixon Ultra EMCCD camera (Andor). The live cells were maintained in glass bottom Petri dishes (MatTek) at 37˚C and 5% $CO_2$ within a microscope incubator (Okolab).

## Software and data analysis

For measurement and counting of the transcription and other signals corresponding to individual FC/DFC units in 3D confocal images, we developed a MatLab based software.[51] The program identifies each unit by creating a maximum intensity projection of the confocal stack and blurring the projection with a Gaussian filter ($\sigma$ = 8–10 pixels), defining the blurred image with a value obtained by Otsu's method for automatic threshold selection. After that, the optical section whereupon the unit had maximum intensity was identified. The final result contains 3D coordinates of each unit, its size (full-width half-maximum), the value of $\chi^2$, and integral intensities in the spheres with radii 1.0, 1.5, 2.0, 2.5, 3.0, 3.5, and 4.0 pixels. The values corresponding to 1.5 pixels seemed to be the most resistant to noise and were used for presentation of the data. FC/DFC units were counted after deconvolution with Huygens software.

For measuring signals in the entire nucleoli we used a custom ImageJ plugin available at https://github.com/vmodrosedem/segmentation-correlation.[45] Based on the confocal stacks, the program identifies the regions occupied by the cell nuclei as well as nucleoli, measures their areas (in pixels), and the intensities, both integral and average, of the signal within these areas.

## Results

### 1. Effects of low temperature on the nucleolar transcription

In the control the incorporated FU is accumulated predominantly in the nucleolar beads which, according to our earlier study,[55] correspond to the FC/DFC units of the nucleoli (Fig 1). The transcription signal in the nucleoplasm appeared as multiple small foci of much lower intensity. After 15 min of incubation at +4˚C (without additional supply of $CO_2$), both HeLa and LECs lost the ability to incorporate 5-fluorouridin (FU). When the cells were returned to the normal conditions (37˚C, 5% $CO_2$), transcription was partly restored in 15 min, and in 30 min the FU incorporation did not visibly differ from the control (Fig 1).

Since incorporation of FU is preceded by its penetration in the cell and phosphorylation, we performed an additional experiment with another RNA predecessor, BrUTP, which was introduced in the HeLa cells by the scratch procedure (Fig 1B).[55, 56] When cells were permeabilized by scratching in the presence of BrUTP for 5 min and then immediately fixed (Fig 1B, left) or washed and incubated for further 10 min at +4˚C (Fig 1B, middle), there was no significant incorporation of the nucleotide. But when the permeabilization was followed by 10 min incubation in the normal conditions (Fig 1B, right), the cells situated along the scratch track displayed the transcription signal in the nucleoli and nucleoplasm. This result confirmed that the transcription was efficiently arrested in our experiments at +4˚C.

It is known that pol I and fibrillarin are particularly sensitive to stress, and their redistribution in the cell nuclei is a common symptom of nucleolar pathology. Therefore, to assess the effect of cold on the FC/DFC units, which are the centers of rDNA transcription and early rRNA processing, we transfected the cells with GFP-RPA43 or GFP-Fibrillarin. At the low

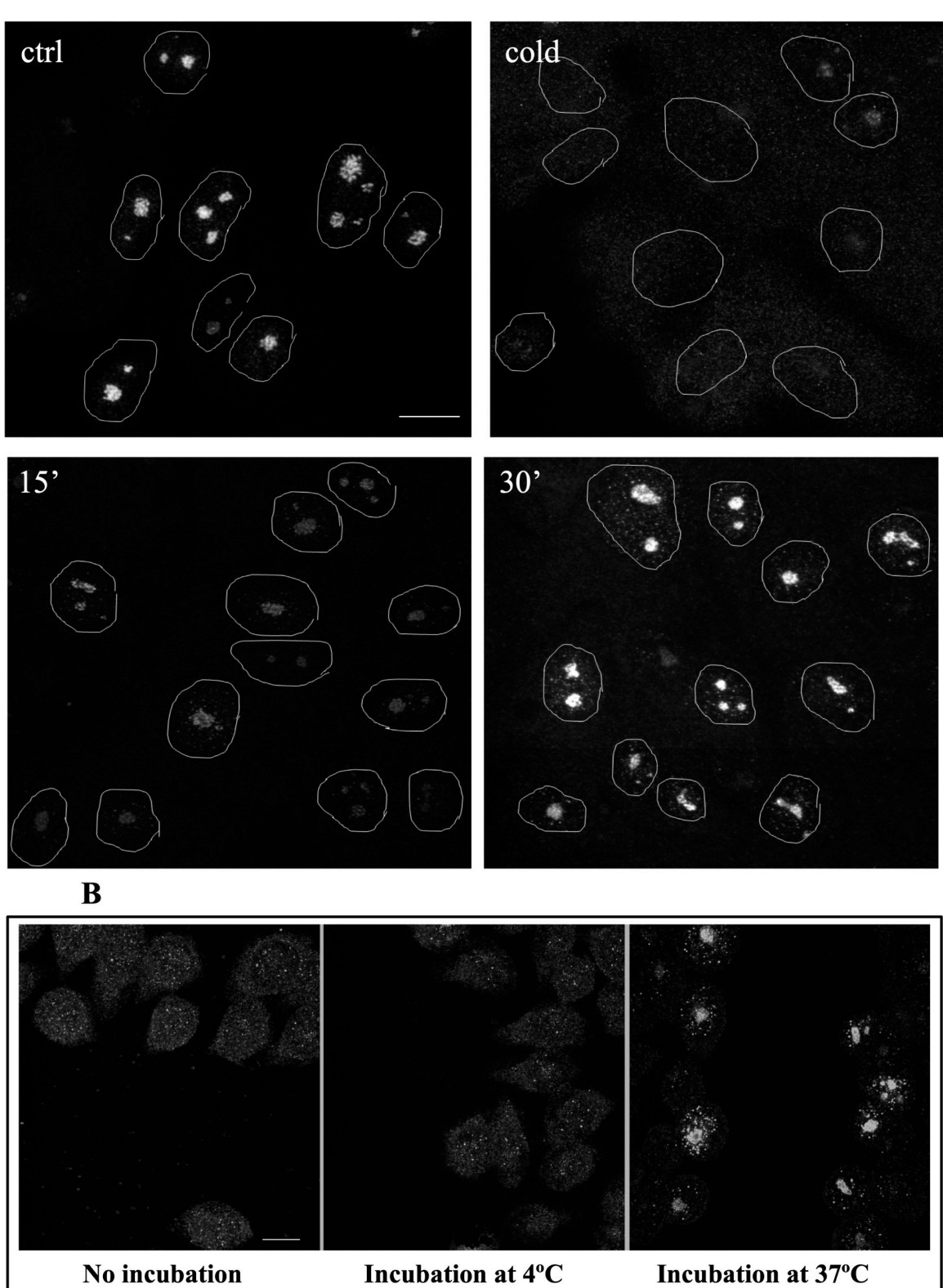

**Fig 1. (A) Transcription in HeLa cells is quickly inhibited at +4˚C and restored at the normal conditions.** The transcription signal (FU incorporation) is accumulated in the nucleoli. The signal disappeared after 15 min of cold treatment (top right); when the cells were transferred to the normal conditions, the signal was partly restored in 15 min and appeared like in the control in 30 min (bottom). **(B) Incorporation of BrUTP.** No significant signal in the cells fixed after the permeabilization immediately (left) or following 10 min of incubation at +4˚C (middle); when permeabilization was followed by 10 min incubation at +37˚C, specifically labelled cells could be observed along the scratch track (right). Scale bar: 10μm.

temperature the GFP-Fibrillarin signal did not change significantly, but the intensity of the RPA43 signal was decreased as average to about 60% of the control level (Fig 2).

Observation of the individual cells also showed that after transferring the cells from the cold to the normal conditions, the intensity of GFP-RPA43 signal in all FC/DFC units increased, although the number of the detectable units did not change (Fig 3).

These experiments show that low temperature causes a quick inhibition of the rDNA transcription, as well as significant though not complete depletion of the pol I pools in the nucleoli.

On the other hand, we observe a quick recovery of the cells without any lasting symptoms of pathology.

## 2. Synchronization of the nucleolar transcription in HeLa and human limbal cells by cold treatment

The experiments described in the previous section indicate that at the low temperature the ribosomal genes are brought to a silent state with a diminished RPA-GFP signal within the FC/DFC units which implies a decreased number of pol I complexes bound to the genes. This synchronization procedure was used for the study of the discontinuous expression of the rDNA in HeLa and LEC cells. Namely, the cells were incubated in cold medium (+4˚C) for 1 h, then transferred to the normal conditions and fixed at different time points from 15 to 150 min with the interval of 15 min. FU was added to the cultivation medium 5 min prior to each fixation. The transcription signal visualized by antibody was then measured in the nucleoli by means of the ImageJ plugin software (see Methods). The results are presented in Fig 4.

In all such experiments the intensity of the transcription signal increased during the first 30 min, then began to decrease. Altogether two cycles of rise and fall have been observed within the period of 150 min, the coefficient of variation (CV) was 0.26. The spectral analysis revealed a significant peak corresponding to the period of 60 min. Since the interval between the measurements was 15 min, the values of the period may be varying from 45 min to 75 min. An additional lower peak at 15 min probably reflected a high frequency noise. In the control, when the cells were kept at 37˚C and fixed at different time points as in the experiment, the fluctuations of the transcription signal intensity were irregular. CV was only 0.07, and the periodogram had two peaks of low amplitude (compare the left and right parts of the Fig 4). In two experiments the period of observation was extended to 210 min, but between 150 and 210 min the fluctuations of the transcription signal appeared irregular with the CV values 0.06, i.e. just like in the control, which indicated that the synchrony in the cell population was lost. These results showed that in HeLa cells the activity of pol I transcription machinery was synchronized by the cold treatment for the period of 150 min, but not longer.

The same experimental procedure was applied to the LECs (Fig 5). In this case the first two cycles were more pronounced and the difference between control and experiment was more significant (compare Fig 5 and Fig 4). Otherwise, the dynamics of the transcription activity after the cold treatment proved to be similar in the studied cell lines. In the LECs, the periodogram had a more distinct peak at 60 min, but the synchronization also did not last longer than 150 min. CV was 0.29, i.e. slightly higher than in HeLa cells. It seems worth mentioning that

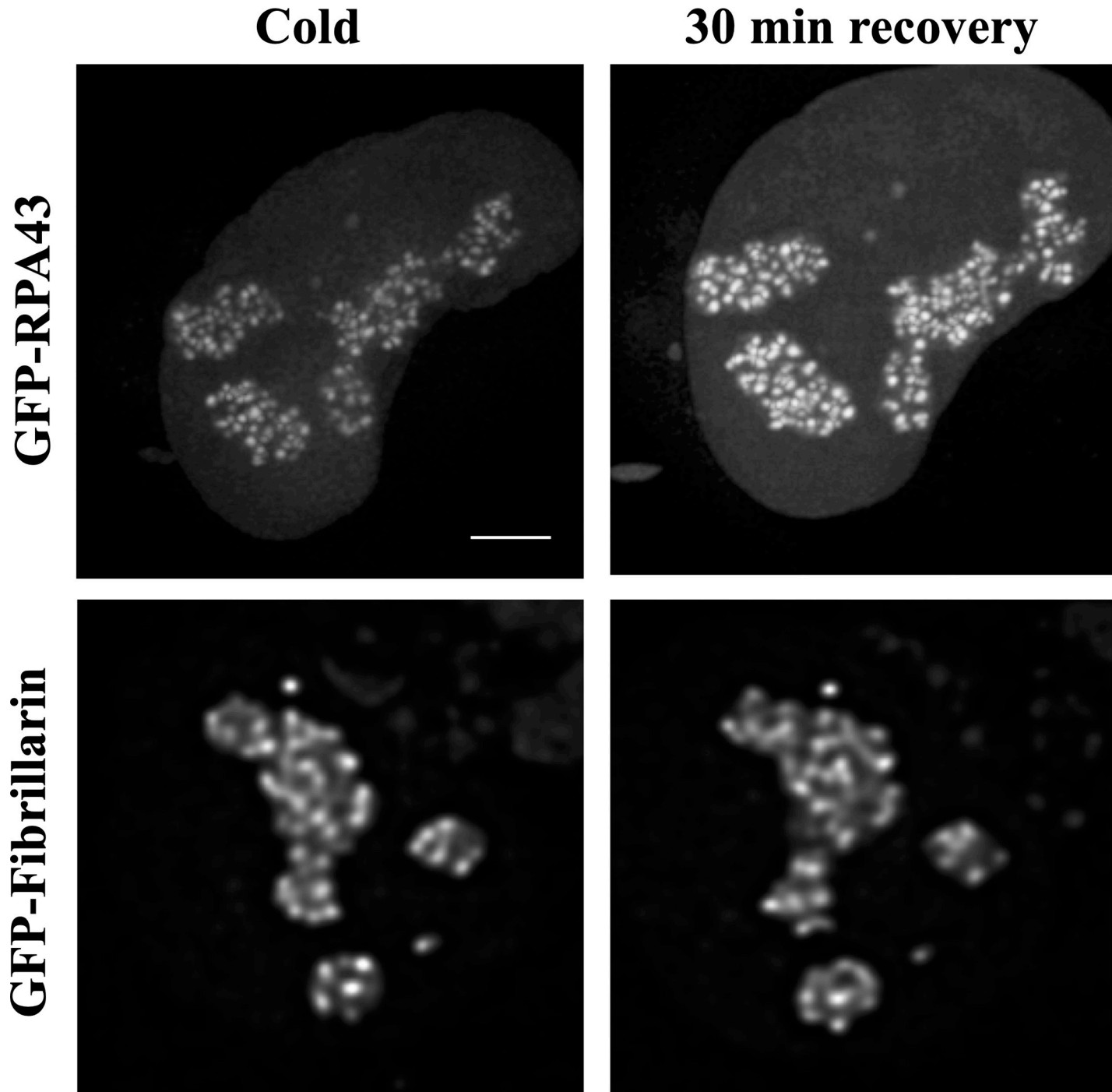

**Fig 2. Following GFP-RPA43 and GFP-Fibrillarin signals in the transfected HeLa cells *in vivo*.** The intensity of the GFP-RPA43 signal is reduced after 15min incubation at +4˚C (left, top) and restored after subsequent 30 min incubation at normal conditions (top, right). The GFP-Fibrillarin signal was not significantly affected by the cooling/warming procedure (bottom). Scale bar: 5μm.

our attempt to synchronize the transcription in human fibroblasts failed, for only a few of these cells recovered quickly enough after the cold treatment.

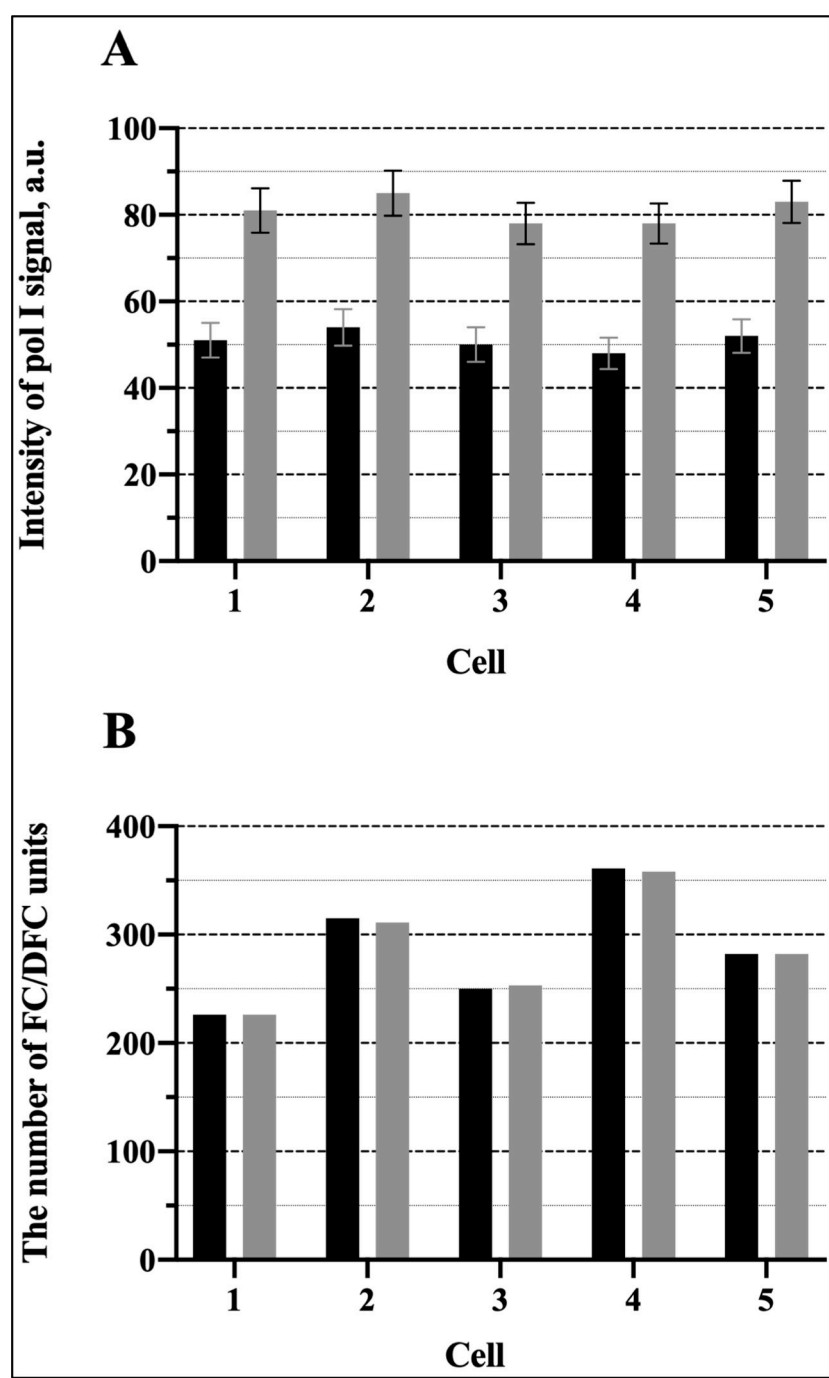

**Fig 3. Effects of cooling/warming (as in Fig 2, top) on the FC/DFC units *in vivo* in the transfected HeLa cells. (A)** intensity of the GFP-RPA43 signal in the individual units after 15min incubation at +4˚C (black columns) and after subsequent 30 min incubation at 37˚C (grey columns). Five cells were observed, and five selected units were followed in each cell. The error bars show SEM. **(B)** the total number of the GFP-RPA43 positive units in five cells after 15min incubation at +4˚C (black columns) and after subsequent 30 min incubation at 37˚C (grey columns). The experiment indicates that at the low temperature pol I escapes from the FC/DFC units.

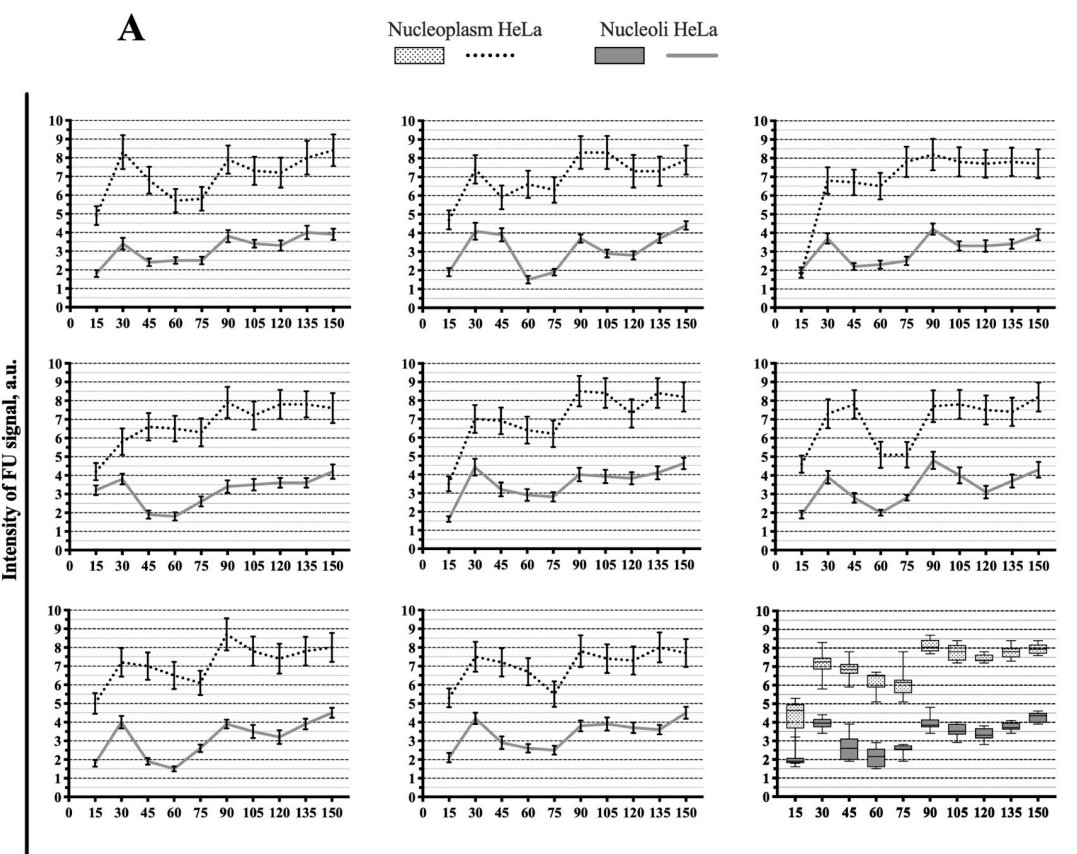

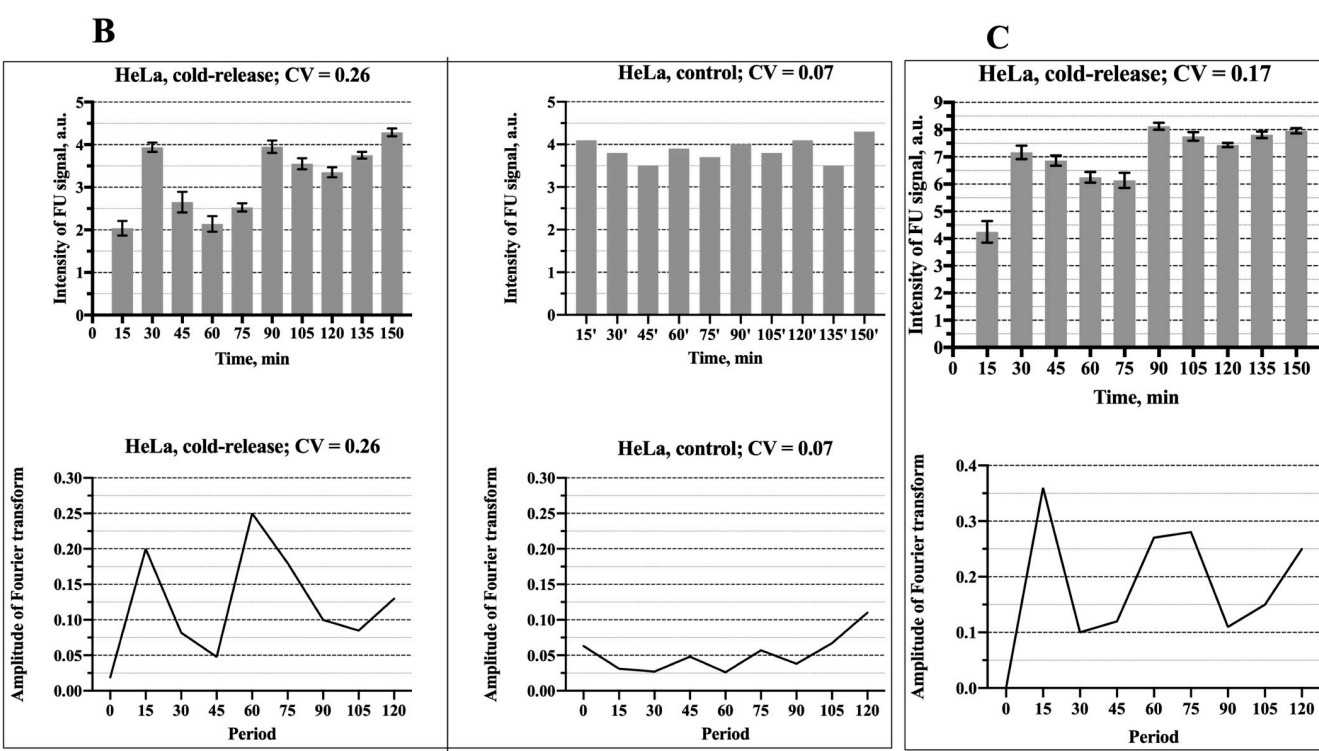

**Fig 4. Fluctuation of the intensity of the transcription signal (incorporated FU) in the whole nucleoli and nucleoplasm of HeLa cells after release from the cold block. (A)** Data of the individual experiments and the box-plot chart. **(B)** Mean values of the transcription signal intensity in the nucleoli after release from the cold block (left, top) and in the nucleoli of the control cells, i.e. without cold treatment (right, top) In the experiment the signal reaches maximal values at 30 min, 90 min, and 150 min. The graph shows mean values obtained from 50 cells in one experiment. Such experiment was repeated 8 times. CV- coefficient of variation. The error bars show SEM. The bottom graphs show the respective periodograms for the experiment (left) and control (right) calculated as amplitudes of the Fourier transforms. The x-axis represents the period (min). **(C)** Mean values of the transcription signal intensity in the nucleoplasm after release from the cold block (top) and the respective periodogram (bottom).

Thus, our experiments indicated that transcription of the ribosomal genes proceeds in a wave-like manner, although the employed synchronization procedure is not equally efficient in various cells.

## 3. Fluctuation of the pol I signal in the cells synchronized by chilling

To confirm our result by an independent set of data, we used chilling shock to synchronize HeLa cells transfected with GFP-RPA-43 (see Fig 6). Measuring the intensity of the pol I signal in the nucleoli of the individual cells, we observed fluctuations similar to those of the transcription signal (Figs 4 and 5). The fluctuations had a relatively low amplitude, and the distinct undulations persisted for no longer than two hours, apparently because only a minor portion of the pol I molecules within FC/DFC units are engaged in the current transcription, as was indicated, for instance, in our earlier work.[51] Nevertheless, the initial increase of the signal intensity during the first 30 min after the cold treatment was followed by a noticeable decrease during the next half hour, which could not be attributed to the effects of recovery. Together with the other results of the present study (Figs 4 and 5) this indicates, that fluctuations of the transcription intensity and pol I levels in the nucleoli are synchronous.

## 4. Synchronization of the transcription in the nucleoplasm by cold treatment

When the LECs or HeLa cells were incubated at +4˚C, the transcription ceased completely in their nucleoplasm as well as in the nucleoli. Measurement of the total FU signal after transferring the cells from the cold to the normal conditions showed symptoms of synchronization: the signal in the nucleoplasm increased for 30 min and then began to decrease (Figs 4A, 4C, 5A and 5C). The average intensity of the transcription signal in the nucleoli and nucleoplasm positively correlated, with the correlation coefficients 0.65 for the HeLa cells and 0.74 for the LECs. But, as one could expect, the total expression of the nucleoplasmic genes was less synchronized. After the initial recovery and subsequent decrease, the signal became rather noisy. The CV was 0.17 and 0.19 in the HeLa and LECs respectively. The periodograms showed a not very distinct peak at 75 min as well as a sharper peak corresponding to higher frequencies. The second peak probably reflects a noisier character of the fluctuations in the nucleoplasm as compared to the nucleoli.

## 5. The FC/DFC units in the course of the transcription fluctuation

Since the measurement of the transcription signal in the whole nucleoli is significantly affected by the fluorescence between the FC/DFC units, we measured the signal also within the units. According to the data presented in the sections 2 and 3 (Figs 4 and 5), the intensity of FU signal in the nucleoli at 15 min and 30 min after the cold treatment may be taken as representatives of the two extreme states of the transcriptional fluctuation in the synchronized cells. Measurement of the FU signal in the individual FC/DFC units of the LECs and HeLa cells using the MatLab based software (see Methods) showed an approximately threefold increase

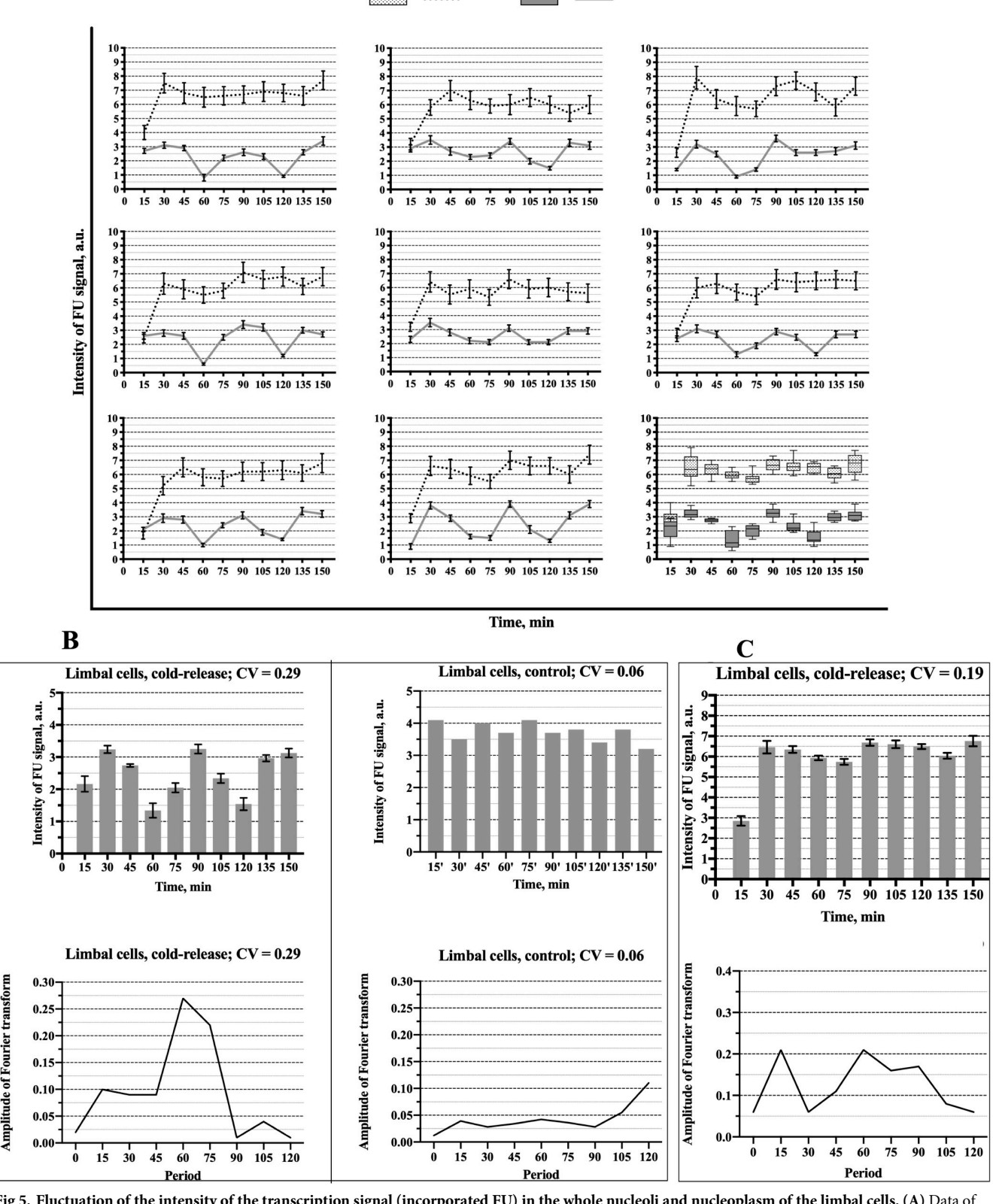

**Fig 5. Fluctuation of the intensity of the transcription signal (incorporated FU) in the whole nucleoli and nucleoplasm of the limbal cells. (A)** Data of the individual experiments and the box-plot chart as in Fig 4A. **(B)** Mean values of the transcription signal intensity in the nucleoli after release from the cold

block (left, top) and in the nucleoli of the control cells (right, top). The figure is analogous to the Fig 4. But in this case, the undulating pattern in the experiment (top, right) is more pronounced, and the periodogram related to the experiment (bottom, left) has a more distinct peak at 60 min. The data are obtained from 8 independent experiments, and in each of them 50 cells were measured. **(C)** Mean values of the transcription signal intensity in the nucleoplasm after release from the cold block (top) and the respective periodogram (bottom).

of the signal intensity between 15 min and 30 min (Fig 7). But the transcription signal never disappeared from the cells completely, so that the average number of the FU-positive FC/DFC units did not change significantly (Fig 7B, right chart).

## Discussion

In our experiments, when the human derived cells were incubated at +4˚C, transcription in their nuclei seemed to be arrested completely (Figs 1 and 7). At the same time the pol I signal in the FC/DFC units of the nucleoli was significantly reduced (Figs 2 and 6), whereas the amount of fibrillarin, which is an essential component of the early rRNA processing, did not change significantly (Fig 2). On the other hand, previous studies, including our own, indicate that the mobile fraction of pol I, apparently responsible for the actual transcription, constitutes less than a half of the entire pool of the enzyme in the units.[51, 54] Therefore, in all probability, the pol I complexes do not "freeze" on their matrices after the arrest of the transcription by the chill shock, but rather detach themselves and escape from the units. After returning to normal conditions, the pools of the enzyme are swiftly restored, and the rRNA synthesis in the cells is synchronized. This effect was used in our work for detection of the pulse-like transcription.

In our previous work we studied the fluctuations of pol I signal, but could not speak about the discontinuous transcription otherwise than hypothetically, since the dynamics of this signal does not necessarily reflect the transcription.[55] Therefore, only after developing the cold/release method of cell synchronization, we obtained the data related to the transcription fluctuations directly.

In thus synchronized HeLa and LEC cells, we observed a wave-like modification of the nucleolar transcription signal with two successive peaks (Figs 4 and 5). It should be mentioned, that the recovery process, which seemed to be limited to the first 30 min after the cold treatment, could not account for the observed dynamics, especially the regularly observed decrease of transcription intensity after the initial increase, as well as more or less distinct second peak. In both kinds of cells, the predominant fluctuation period estimated by the spectral analysis was about 60 min. A similar value of the period was obtained in our previous work for the fluctuations of the GFP-RPA43 signal.[51]. After the two distinct cycles, the waves were damped; probably because of their irregularity and variability in the individual cells. Nevertheless, our data indicate that the ribosomal genes are expressed discontinuously, with intervals of 45–75 min between the bursts.

In our review on the discontinuous transcription, we indicated what seemed to be four main patterns in which this phenomenon may be manifested: the typical busts; the undulating pattern; the regular pulsing; and the rare transcription events.[1] As mentioned above, the fluctuations observed in our study do not seem to belong to the regular type. Rare events also must be excluded, since rDNA transcription is very intensive throughout the entire interphase. The typical bursts are separated by the relatively long periods of silence. But we observed no diminishing of the number of FU positive (Fig 7B) or pol I positive (Fig 3) FC/DFC units in the course of the experiment, although the mean intensity of the incorporated FU signal in the individual units was greatly reduced at the points of minimal transcription activity (Fig 7B).

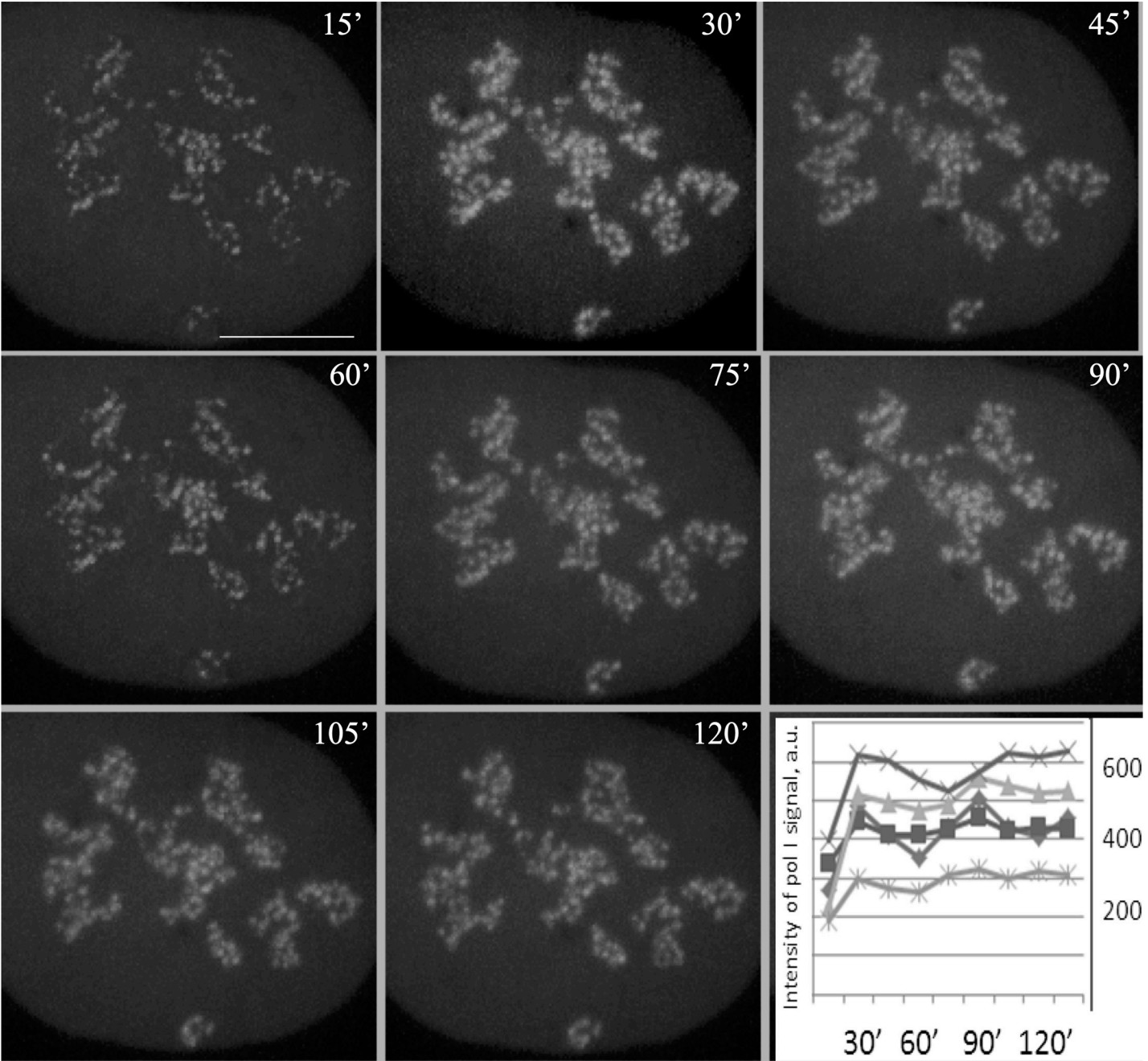

**Fig 6. Fluctuation of the intensity of GFP-RPA43 signal in the nucleoli of HeLa cells after the cold treatment.** The eight successive images of the same transfected cell photographed every 15 min after the release from the cold block. The intensity of the signal at 15 min as well as at 60 min after the release is visibly lower than at other points. The graph at the bottom right shows records of the pol I signal intensity in five cells at different time points after the release. Each curve represents one cell. All curves have two peaks at 30 min and at 90 min or close to it, as in the case of FU incorporation (Figs 4 and 5). Scale bar: 5μm.

Therefore, the observed fluctuation of rDNA transcription most likely belongs to the undulating pattern, in which the bursts are alternated by periods of relatively rare transcription events.

Additionally, our method of synchronization allowed us to obtain averaged data concerning the fluctuations in the nucleoplasmic genes, since their expression was also inhibited by the cold treatment. After this procedure, the total transcription signal in the nucleoplasm

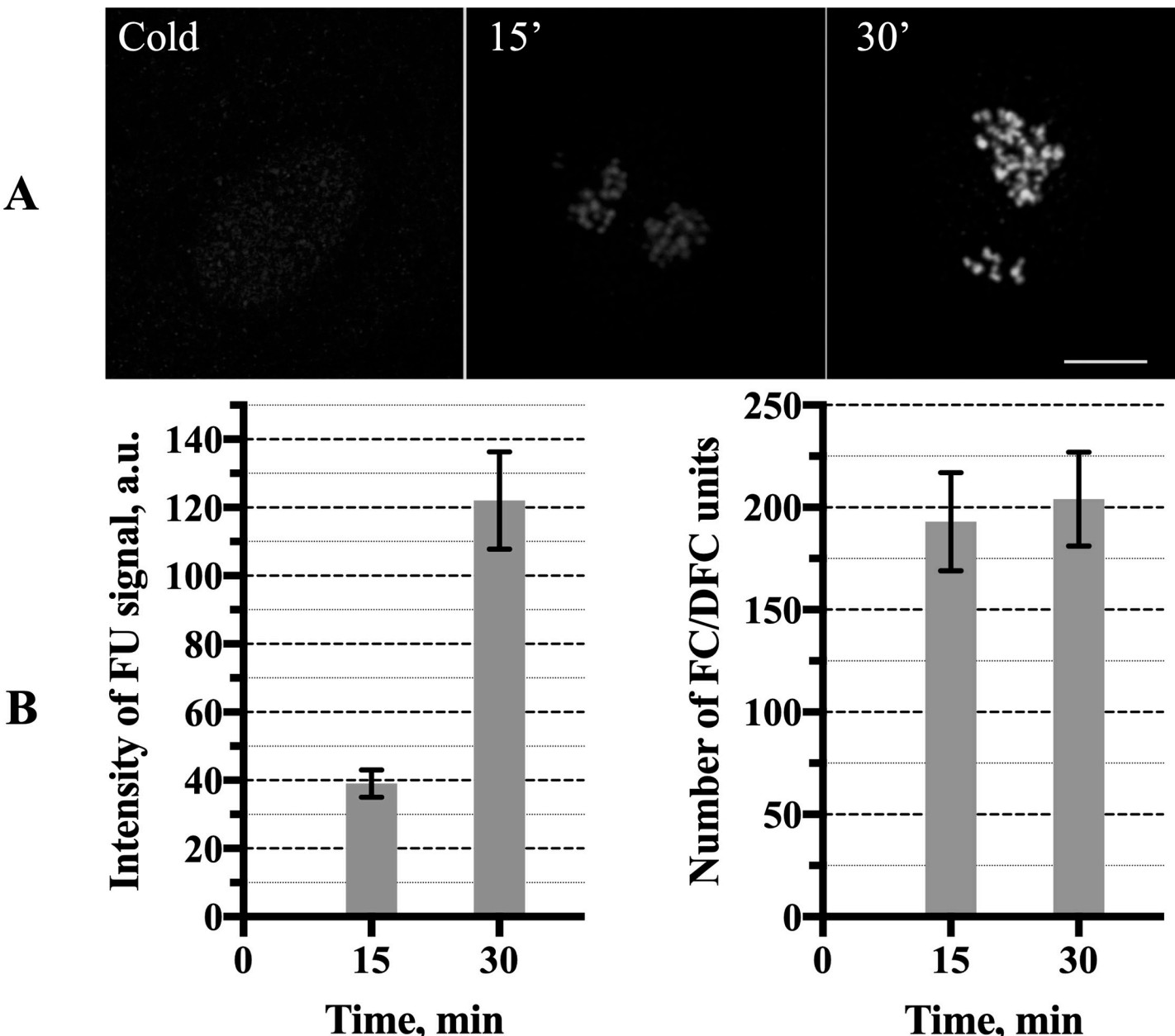

**Fig 7. The transcription signal (incorporated FU) in the FC/DFC units of the limbal epithelial cells after the cold treatment.** (A) no FU incorporation in the cells incubated at +4°C for 15 min (left); a weak FU signal in the cell incubated at 37°C for 15 min after the chilling (middle); 30 min; completely recovered FU signal in the cell incubated at 37°C for 30 min after the chilling (right). Scale bar: 5 μm. (B) the average (from 50 cells) intensity of the FU signal in the individual FC/DFC units measured in the cells incubated at 37°C for 15 min and 30 min after the chilling. The increase is statistically significant (P < 0.0001, according to the Student's t-test) (left). The average (from 50 cells) number of the FU positive FC/DFC units in the cells incubated for 15 min and 30 min at +37°C after the cold treatment; the differences are statistically insignificant (right). The data show an initial quick recovery of the transcription in the units without changing their number (see Fig 5).

showed symptoms of fluctuations with two discernible, though not very distinct, peaks (Figs 4A, 4C, 5A and 5C). Evaluating these results, we have to keep in mind that various nucleoplasmic genes in the same cell display a wide range of transcriptional kinetic behavior (reviewed in Smirnov et al. [1]).[4, 10, 57, 58] Moreover, some of these genes are expressed in typical bursts with long periods of silence, during which they cannot be detected by FU incorporation. We should also mention that the status of the nucleoplasmic RNA polymerases at the low temperature

was not examined in our experiments, and thus we do not know how efficiently the transcription was synchronized. Nevertheless, the presence of two significant peaks on the periodograms (Figs 4C and 5C) suggests that numerous genes in the nucleoplasm were transcribed in a pulse-like manner with periods close to 15 min and 75 min.

Thus, our results indicate that ribosomal genes in human cells are expressed discontinuously, and their transcription follows undulating pattern with predominant period of about 60 min.

## Acknowledgments

The work was supported by research project BBMRI_CZ LM2018125, European Regional Development Fund, project EF16_013/0001674, by the Grant Agency of Czech Republic (19-21715S) and by Charles University (Progres Q25 and Q28). SK acknowledges the financial support from the Czech Science Foundation Grant No. 1825144Y. The funders had no role in study design, data collection and analysis, decision to publish, or preparation of the manuscript.

## Author Contributions

**Conceptualization:** Evgeny Smirnov.

**Data curation:** Peter Trosan, Pavel Studeny, Katerina Jirsova.

**Formal analysis:** Evgeny Smirnov, Joao Victor Cabral, Sami Kereïche.

**Investigation:** Evgeny Smirnov, Joao Victor Cabral, Katerina Jirsova, Dušan Cmarko.

**Methodology:** Evgeny Smirnov, Peter Trosan.

**Software:** Sami Kereïche.

**Supervision:** Evgeny Smirnov, Dušan Cmarko.

**Validation:** Evgeny Smirnov, Joao Victor Cabral.

**Writing – original draft:** Evgeny Smirnov.

**Writing – review & editing:** Peter Trosan, Joao Victor Cabral, Pavel Studeny, Sami Kereïche, Katerina Jirsova, Dušan Cmarko.

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
