## [Decision Letter · Decision Letter 0]

24 Oct 2019

PONE-D-19-25095

Discontinuous transcription of ribosomal DNA in human cells

PLOS ONE

Dear Dr. Smirnov,

Thank you for submitting your manuscript to PLOS ONE. After careful consideration, we feel that it has merit but does not fully meet PLOS ONE’s publication criteria as it currently stands. Therefore, we invite you to submit a revised version of the manuscript that addresses all the points raised during the review process.

In particular:

1. your data would benefit of better presentation including laid out rationales for the experiments/choice of experimental approaches (see Rev 1), presentation of individual data points (as suggested by Rev 2), extended discussion (including clarifying the issue of novelty as identified by Rev 3 as well as stress recovery effects- Rev 2), fixed issues with figure numberings, abbreviations, full descriptions of statistics.

2. Confirming findings with 5FU using another approach is critical for interpretation of your current data (see comments by Rev 2 and Rev 3)  

We would appreciate receiving your revised manuscript by 60 days from the date of this decision letter. To enhance the reproducibility of your results, we recommend that if applicable you deposit your laboratory protocols in protocols.io, where a protocol can be assigned its own identifier (DOI) such that it can be cited independently in the future. For instructions see: http://journals.plos.org/plosone/s/submission-guidelines#loc-laboratory-protocols

We look forward to receiving your revised manuscript.

Kind regards,

Michal Hetman

Academic Editor

PLOS ONE

**Journal Requirements:**

2. Thank you for including your ethics statement "The study followed the standards of the Ethics Committees of the General Teaching Hospital and the First Faculty of Medicine of Charles University, Prague, Czech Republic (Ethics Committee of General Univeristy Hospital, Prague approval no. 8/14 held on January 23, 2014), and adhered to the tenets set out in the Declaration of Helsinki. We obtained human cadaver corneoscleral rims from 10 donors, which were surplus from surgery and stored in Eusol-C (Alchimia, Padova, Italy), from the Department of Ophthalmology, General University Hospital in Prague, Czech Republic, for the study. On the use of the corneoscleral rims, based on Czech legislation on specific health services (Law Act No. 372/2011 Coll.), informed consent is not required if the presented data are anonymized in the form."

a) Please amend your current ethics statement to confirm that your named institutional review board or ethics committee specifically approved this study.

For additional information about PLOS ONE ethical requirements for human subjects research, please refer to " ext-link-type="uri" xlink:type="simple">http://journals.plos.org/plosone/s/submission-guidelines#loc-human-subjects-research."

**Comments to the Author**

1. Is the manuscript technically sound, and do the data support the conclusions?

Reviewer #1: Yes

Reviewer #2: Partly

Reviewer #3: Partly

2. Has the statistical analysis been performed appropriately and rigorously? 

Reviewer #1: Yes

Reviewer #2: No

Reviewer #3: No

3. Have the authors made all data underlying the findings in their manuscript fully available?

Reviewer #1: Yes

Reviewer #2: No

Reviewer #3: No

4. Is the manuscript presented in an intelligible fashion and written in standard English?

Reviewer #1: Yes

Reviewer #2: Yes

Reviewer #3: Yes

5. Review Comments to the Author

Reviewer #1: The paper by Eugene Smirnov et al. describes the analysis of rDNA expression during about 2 h after release of human cells from cold treatment. FU incorporation was reduced by the cold treatment and the signal reached the normal level after 30 min of incubation at 37°C.

Using transfected plasmid carrying Pol I gene fused with GFP or GFP-Fibrillarin construct they observed that cold treatment reduced the levels of Pol I in the FC/DFC units. In subsequent experiments they demonstrated that there are fluctuations of incorporated FU in the whole

nucleoli during 120-150 min.

The experimental data confirm the conclusion that rDNA genes are transcribed discontinuously during short period after incubation at 37°C.

The paper should be re-written. Authors should first describe all experiments in the text in more detail. E.g., why they use this particular experiment in the text. why they use the transfections with Pol I gene and Fibrillarin gene. How they measure the intensity and amplitude. What are a.u. and a.e. units? What secondary Ab were used in FU detection? Each panel in the Figure should be described.

Reviewer #2: The work describes the study of discontinuous expression of rRNA genes. This manuscript builds on their previous work published in the Nucleus journal. In a new work, the authors investigated discontinuous expression of ribosomal genes after inhibition of transcription using cold stress. It seems that the data presented are not enough to convincingly illustrate the assumptions made.

1. The main emphasis in the work is made on the analysis of the intensity of label (FU) incorporation into the nucleoli of cultured cells. However, the inclusion of FU depends not only on the intensity of expression, but also on the rate of its cellular uptake and phosphorylation in the cell. Therefore, confirmation of this key result by an independent method is necessary. The authors show in Fig. 2 and 3A (and in their previous work, this was also demonstrated) that the nucleolur accumulation of RPA 43 also changes with time. It is possible to confirm the presence of waves of intensity of ribosomal RNA expression using this method. The live cell imaging of RPA 43 during during the recovery of cells after cold stress (with estimations and statistical analysis) can illustrate described waves more accurately than the estimation of FU incorporation for overall cell population.

2. The authors indicate that the data on fluctuations and transcription cycles were obtained by averaging over several experiments (8 repetitions). It seems that it is necessary to present not only the result of averaging over 8 experiments, but also the curves for all individual experiments (plus, the result of averaging). To appreciate how is the distribution of averaged values among the time points, instead of using bar graphs the data should be represented using box-plots with the individual values as dots. Also, the statistical analysis of detected fluorescence intensity fluctuations should be presented. These changes were statistically significant or not?

3. I think that the experimental model used (restoration of transcription after cold stress) cannot be interpreted as synchronization. For human cells, this is very severe stress. And it seems that the subsequent processes should be interpreted as a process of recovery from stress. In this case, the transcription fluctuations can be connected with the process of cell restoration, which may differ from fluctuations in the control culture (which was described in the article in the Nucleus journal). Authors should at least briefly discuss such an interpretation of their data.

Reviewer #3: The main goal of the study by Smirnov et al. “Discontinuous transcription of ribosomal DNA in human cells” was to evaluate nucleolar and nucleoplasmic transcription in two types of human cells (HeLa and epithelial limbal cells) after their synchronization with a low (4o C) temperature followed be the release from the cold shock as compared with untreated controls. Aims were reached by incubation of cells with 5-fluorouridine as precursor of RNA synthesis, expression of plasmids encoding RNA pol I subunit (RPA43) and fibrillarin fused with GFP. The intensity and number of signals were examined using a MatLab based software [ref 51] and Image J facilities. The authors concluded that: (1) chilling of cells results in arrest of pol I (nucleolar) and pol II (nucleoplasmic) transcription but does not displace all pol I complexes from their intrinsic locations; (2) cell release from the chilling conditions restores rDNA transcription to control values; (3) the restoration of rDNA transcription follows a wave-like manner (within 15-210 min of observation) and is discontinuous process.

Comments: The major question concerns the principal novelty of the reviewed paper as compared with the papers recently published by the same authors in “Nucleus” (M. Hornáček et al., Fluctuations of pol I and fibrillarin contents of the nucleoli. Nucleus, 2017, 8: 421-432; Smirnov et al., Discontinuous transcription Nucleus, 2018, 9: 149-160). In both publications, it is stated that ribosomal genes, like other genes, are transcribed in pulse-like manner (e.g., Hornáček et al., 2018, page 150), while it is well known that rRNA genes are transcribed during the entire cell cycle [ref. 49], which duration is much longer (roughly 24 hours) than the duration of observations in the current study (150-210 min).

Minor questions:

Lines 25, 89 “in the populations of tumour derived” – Unclear meaning

Line 92: Methods

Conditions for cell chilling should be described in this section instead of Results.

Lines 159, 160 (Fig 1) – It is unclear, where are the nuclear boundaries, and how the authors determined that “… FU is accumulated predominantly in the FC/DFC units of the nucleoli” without using any markers for FC/DFC?

Lines 187, 190 (legend for Figure 3): “bars” should apparently be replaced by “columns”.

Fig.3A: What are the vertical bars: SEM or standard deviation (Ϭ).

Fig. 3B: SEMs (or Ϭ) are not indicated.

Lines 193, 194: It is remained unspecified how the authors distinguish between negative labeling of cells with 5-FU (5-fluorouridine) caused by inhibition of rDNA transcription from non-penetration of the precursor in cells in cold conditions. In addition, before incorporation in nascent pre-rRNA FU must be bound to ATP and this process most likely is suppressed by a low temperature. By other words, 5-FU was unincorporated not because genes were not transcribed, but because the precursor was inaccessible to nascent RNAs upon cold conditions.

Fig. 4 (Lines 207-213).

A – there are no SEM (or Ϭ) on the columns and therefore it is impossible to compare differences between various time-points statistically. The latter makes the authors statement about a fluctuating manner of rDNA transcription uncertain.

B – on the periodograms, the horizontal axis scale does not correspond to the relative graphs in Fig. 4A.

Fig. 5 (Lines 241-246): See comments to Fig. 4.

Fig. 6 (Lines 264-267): See comments to Fig. 4.

The images below Fig. 7 are not described (are they copies?).

Unfortunately, figures are not numbered that makes their identification complicated.

6. PLOS authors have the option to publish the peer review history of their article (what does this mean?). If published, this will include your full peer review and any attached files.

Reviewer #1: No

Reviewer #2: No

Reviewer #3: No

---

## [Author Response · Author response to Decision Letter 0]

6 Dec 2019

Our specific replies to the reviewers are as follows (here the remarks requiring our reply are put in bold font):

Reviewer #1: The paper by Eugene Smirnov et al. describes the analysis of rDNA expression during about 2 h after release of human cells from cold treatment. FU incorporation was reduced by the cold treatment and the signal reached the normal level after 30 min of incubation at 37°C. Using transfected plasmid carrying Pol I gene fused with GFP or GFP-Fibrillarin construct they observed that cold treatment reduced the levels of Pol I in the FC/DFC units. In subsequent experiments they demonstrated that there are fluctuations of incorporated FU in the whole nucleoli during 120-150 min.The experimental data confirm the conclusion that rDNA genes are transcribed discontinuously during short period after incubation at 37°C.

The paper should be re-written. Authors should first describe all experiments in the text in more detail. E.g., why they use this particular experiment in the text, why they use the transfections with Pol I gene and Fibrillarin gene.

In the revised version of the text we furnished our description with further details. In particular, we observed that, since redistribution of pol I and fibrillarin in the cell nuclei is a common symptom of nucleolar pathology, the study of cells transfected with GFP-RPA43 or GFP-Fibrillarin was needed to assess the reaction of the cells to the chilling (Results, 1st section, 3rd paragraph). 

How they measure the intensity and amplitude. What are a.u. and a.e. units? 

As in our previous works, we measured the signal intensity in arbitrary units (a.u.), but “a.e.“ in Fig 7 was an error, which we corrected in the revised text. “Amplitude” refers to the amplitude of the Fourier transform, i.e. the value of the periodogram. This was mentioned in the legend to the Fig 4 of our original manuscript. 

What secondary Ab were used in FU detection? 

We indicated the antibody in the revised Methods.

Each panel in the Figure should be described.

We extended and corrected the description of the Figures.

Reviewer #2: The work describes the study of discontinuous expression of rRNA genes. This manuscript builds on their previous work published in the Nucleus journal. In a new work, the authors investigated discontinuous expression of ribosomal genes after inhibition of transcription using cold stress. It seems that the data presented are not enough to convincingly illustrate the assumptions made.

1. The main emphasis in the work is made on the analysis of the intensity of label (FU) incorporation into the nucleoli of cultured cells. However, the inclusion of FU depends not only on the intensity of expression, but also on the rate of its cellular uptake and phosphorylation in the cell. Therefore, confirmation of this key result by an independent method is necessary. The authors show in Fig. 2 and 3A (and in their previous work, this was also demonstrated) that the nucleolar accumulation of RPA 43 also changes with time. It is possible to confirm the presence of waves of intensity of ribosomal RNA expression using this method. The live cell imaging of RPA 43 during during the recovery of cells after cold stress (with estimations and statistical analysis) can illustrate described waves more accurately than the estimation of FU incorporation for overall cell population.

We did the confirming experiments on the GFP-RPA43 transformed cells, as suggested by the reviewer. The results are described in the text (Results, section 4) and shown in the Fig 6 of the revised manuscript. 

2. The authors indicate that the data on fluctuations and transcription cycles were obtained by averaging over several experiments (8 repetitions). It seems that it is necessary to present not only the result of averaging over 8 experiments, but also the curves for all individual experiments (plus, the result of averaging). 

In the revised manuscript we show the curves corresponding to individual experiments as well as the result of the averaging with the errors (Fig 4B,C and 5B,C of the revised manuscript). 

To appreciate how is the distribution of averaged values among the time points, instead of using bar graphs the data should be represented using box-plots with the individual values as dots. 

We created the box-plots, as suggested by the reviewer (Fig 4A and 5A of the revised manuscript).

Also, the statistical analysis of detected fluorescence intensity fluctuations should be presented. These changes were statistically significant or not?

We added statistical data (significance levels according to the Student’s t-criterion) showing significant change in the intensity of the FU signal and insignificant difference in the number of FC/DFC units between 15 min and 30 min after the cold treatment (see legend to the Fig 7) 

3. I think that the experimental model used (restoration of transcription after cold stress) cannot be interpreted as synchronization. For human cells, this is very severe stress. And it seems that the subsequent processes should be interpreted as a process of recovery from stress. In this case, the transcription fluctuations can be connected with the process of cell restoration, which may differ from fluctuations in the control culture (which was described in the article in the Nucleus journal). Authors should at least briefly discuss such an interpretation of their data.

We provide a brief discussion suggested by the reviewer (see Results, 1st section, last paragraph and Discussion, 2nd paragraph). There was indeed a recovery process, but it seemed to be limited to the first 30 min after the chilling, and it could not account for the regularly observed decrease of transcription intensity at 60 and 120 min (see Fig 4 and 5). 

We would also observe that chilling does not seem to be a severe stress for the cultivated cells. Unlike the common methods of transcription inhibition (such as actinomycin, amanitin or DRB treatment), it causes no lasting abnormalities; in fact, we did not observe any changes in the nucleolar distribution of fibrillarin after the cold treatment (see Fig 2). 

Relative mildness of the chilling shock may also be seen from the fact that cells are well preserved when kept outside the incubator (e.g. during transportation) at low temperature (without freezing). 

Reviewer #3: The main goal of the study by Smirnov et al. “Discontinuous transcription of ribosomal DNA in human cells” was to evaluate nucleolar and nucleoplasmic transcription in two types of human cells (HeLa and epithelial limbal cells) after their synchronization with a low (4o C) temperature followed be the release from the cold shock as compared with untreated controls. Aims were reached by incubation of cells with 5-fluorouridine as precursor of RNA synthesis, expression of plasmids encoding RNA pol I subunit (RPA43) and fibrillarin fused with GFP. The intensity and number of signals were examined using a MatLab based software [ref 51] and Image J facilities. The authors concluded that: (1) chilling of cells results in arrest of pol I (nucleolar) and pol II (nucleoplasmic) transcription but does not displace all pol I complexes from their intrinsic locations; (2) cell release from the chilling conditions restores rDNA transcription to control values; (3) the restoration of rDNA transcription follows a wave-like manner (within 15-210 min of observation) and is discontinuous process.

Comments: The major question concerns the principal novelty of the reviewed paper as compared with the papers recently published by the same authors in “Nucleus” (M. Hornáček et al., Fluctuations of pol I and fibrillarin contents of the nucleoli. Nucleus, 2017, 8: 421-432; Smirnov et al., Discontinuous transcription Nucleus, 2018, 9: 149-160). In both publications, it is stated that ribosomal genes, like other genes, are transcribed in pulse-like manner (e.g., Hornáček et al., 2018, page 150), while it is well known that rRNA genes are transcribed during the entire cell cycle [ref. 49], which duration is much longer (roughly 24 hours) than the duration of observations in the current study (150-210 min).

In the quoted work (Hornáček et al., 2017) we studied the fluctuations of pol I signal, but could not speak about transcription otherwise than hypothetically, since, for instance, our FRAP experiments indicated that most of the pol I molecules within FC/DFC units were not engaged in the current transcription. Only after developing the cold/release method of cell synchronization, we obtained the data related to the transcription directly. Hence, the straightforward title of our new study.

In the revised text we emphasized the novelty of the present work (Discussion, 2nd paragraph).

Minor questions:

Lines 25, 89 “in the populations of tumour derived” – Unclear meaning

We changed “tumour derived“ to “HeLa“ 

Line 92: Methods

Conditions for cell chilling should be described in this section instead of Results.

We added this description to the Methods, but left it also in the Results for the convenience of the reader

Lines 159, 160 (Fig 1) – It is unclear, where are the nuclear boundaries, 

In the revised manuscript, we showed the outlines of the nuclei on the indicated Figure. 

and how the authors determined that “… FU is accumulated predominantly in the FC/DFC units of the nucleoli” without using any markers for FC/DFC?

In the revised manuscript we provided the reference (Results, 1st section, 1st paragraph) to our previous publication (Smirnov et al, 2016), where we demonstrated colocalization of FU and pol I fluorescent signals, as well as correspondence of the latter to electron microscopic images of the FC/DFC units. 

Lines 187, 190 (legend for Figure 3): “bars” should apparently be replaced by “columns”.

We made the correction.

Fig.3A: What are the vertical bars: SEM or standard deviation (Ϭ).

The error bars signified SEM. We indicated this in the legends of the revised manuscript.

Fig. 3B: SEMs (or Ϭ) are not indicated.

As the legend says, the Fig 3B represents the total numbers of the FC/DFC units in the five cells (unlike Fig 3A, which represents the average intensities). Thus there are no statistical errors in this case. 

Lines 193, 194: It is remained unspecified how the authors distinguish between negative labeling of cells with 5-FU (5-fluorouridine) caused by inhibition of rDNA transcription from non-penetration of the precursor in cells in cold conditions. In addition, before incorporation in nascent pre-rRNA FU must be bound to ATP and this process most likely is suppressed by a low temperature. By other words, 5-FU was unincorporated not because genes were not transcribed, but because the precursor was inaccessible to nascent RNAs upon cold conditions.

In the revised version of the manuscript, we present the data of a new experiment, in which BrUTP was used instead of FU (See Results, 1st section, Fig 1B and Methods for the experiment description). The results seemed to show convincingly that cold alone inhibited the transcription in our experiments. 

Fig. 4 (Lines 207-213). A – there are no SEM (or Ϭ) on the columns and therefore it is impossible to compare differences between various time-points statistically. The latter makes the authors statement about a fluctuating manner of rDNA transcription uncertain.

Fig. 5 (Lines 241-246): See comments to Fig. 4.

Fig. 6 (Lines 264-267): See comments to Fig. 4.

Combining the suggestions of the reviewers 2 and 3, we provided the error bars, as well as the graphs corresponding to individual experiments and box-plots in the Figs 4,5.

B – on the periodograms, the horizontal axis scale does not correspond to the relative graphs in Fig. 4A.

Our error was in applying the same term “time period” to different variables. In the revised text, to distinguish between the real time of the experiment and the parameter of the periodogram, we changed the first to ”Time“ and the second to ”Period.“

The images below Fig. 7 are not described (are they copies?).

Indeed, the images appeared in the pdf version as the result of an error.

Unfortunately, figures are not numbered that makes their identification complicated.

We added the numbers.

---

## [Decision Letter · Decision Letter 1]

7 Jan 2020

PONE-D-19-25095R1

Discontinuous transcription of ribosomal DNA in human cells

PLOS ONE

Dear Dr. Smirnov,

Thank you for submitting your manuscript to PLOS ONE. After careful consideration, we feel that it has merit but does not fully meet PLOS ONE’s publication criteria as it currently stands. Therefore, we invite you to submit a revised version of the manuscript that addresses the points raised during the review process.

Those include: 

-clarifying statistical analysis results as pointed by Rev#2

-reorganizing the abstract to reflect the revised content accurately (as pointed by Rev#3)

-clarifying the methods section as suggested by Rev#3

We would appreciate receiving your revised manuscript in 30 days from the date of this decision letter. To enhance the reproducibility of your results, we recommend that if applicable you deposit your laboratory protocols in protocols.io, where a protocol can be assigned its own identifier (DOI) such that it can be cited independently in the future. For instructions see: http://journals.plos.org/plosone/s/submission-guidelines#loc-laboratory-protocols

We look forward to receiving your revised manuscript.

Kind regards,

Michal Hetman

Academic Editor

PLOS ONE

Reviewers' comments:

Reviewer's Responses to Questions

**Comments to the Author**

1. If the authors have adequately addressed your comments raised in a previous round of review and you feel that this manuscript is now acceptable for publication, you may indicate that here to bypass the “Comments to the Author” section, enter your conflict of interest statement in the “Confidential to Editor” section, and submit your "Accept" recommendation.

Reviewer #1: All comments have been addressed

Reviewer #2: All comments have been addressed

Reviewer #3: (No Response)

2. Is the manuscript technically sound, and do the data support the conclusions?

Reviewer #1: Yes

Reviewer #2: Yes

Reviewer #3: Partly

3. Has the statistical analysis been performed appropriately and rigorously? 

Reviewer #1: Yes

Reviewer #2: No

Reviewer #3: Yes

4. Have the authors made all data underlying the findings in their manuscript fully available?

Reviewer #1: Yes

Reviewer #2: Yes

Reviewer #3: No

5. Is the manuscript presented in an intelligible fashion and written in standard English?

Reviewer #1: Yes

Reviewer #2: Yes

Reviewer #3: Yes

6. Review Comments to the Author

Reviewer #1: Synchronized HeLa and LEC cells demonstrated two successive peaks of rDNA transcription. These picks are now presented for individual cells and for averaging.

The text and Legends to Figures in the revised paper include the details that help a reader to understand the paper.

All my concerns were addressed.

Reviewer #2: Minor questions

1. Line 336. The usual p-value in t-test is 0.05.

2. The data in Fig. 4A and 5A. Is each curve based on a measurement of 50 cells? Maybe then it is worth presenting for each point the data not only on the mean, but also the standard deviation?

3. The new figures are not embedded in the logic of the article. In the current version, the last panel with box-plot in fig. 4a and fig. 4b (+ last panel with box-plot in 5a and fig. 5b) duplicate each other.

4. It is not clear why the error bars for the control cells are not provided (Fig. 4B and Fig. 5B, right panels)?

Reviewer #3: Comments to the revised version of the paper by Smirnov et al. “Discontinuous transcription of ribosomal DNA in human cells”

In general, the authors took into account my comments and adequately answered the questions.

However:

(1) The Abstract remains almost unchanged. For example, it does not include mentioning the results of BrUTP experiments, which were included to the revised paper (lines 176-183). According to my opinion, these experiments are crucially important for interpretation of the FU-labeling data. The same is also true for the last paragraph of Introduction, where the authors summarize the main paper results.

(2) Methods

It remains unclear, why the BrUTP labeling protocol is described only “briefly” (line 135). The publications cited by the authors for the scratch procedure used in the current study [55, 56] apply cell labeling with DNA replication but not with any transcription precursors.

What was the duration of cell fixation with methanol (line 133)?

What was the working concentration of BrUTP?

Line 140: “cy3-conjugated” should be replaced by “Cy3-conjugated”.

7. PLOS authors have the option to publish the peer review history of their article (what does this mean?). If published, this will include your full peer review and any attached files.

Reviewer #1: No

Reviewer #2: No

Reviewer #3: No

---

## [Author Response · Author response to Decision Letter 1]

22 Jan 2020

Reviewer #1: 

All my concerns were addressed.

Reviewer #2: Minor questions

1. Line 336. The usual p-value in t-test is 0.05.

We presented the result in the form suggested by the reviewer.

2. The data in Fig. 4A and 5A. Is each curve based on a measurement of 50 cells? Maybe then it is worth presenting for each point the data not only on the mean, but also the standard deviation?

We showed the deviations on the indicated graphs

3. The new figures are not embedded in the logic of the article. In the current version, the last panel with box-plot in fig. 4a and fig. 4b (+ last panel with box-plot in 5a and fig. 5b) duplicate each other.

The figure 4B contains mean values with standard deviations, whereas Fig 4A shows the extreme values of the set, the median and two other quartiles. We think that both indicated figures are useful.

4. It is not clear why the error bars for the control cells are not provided (Fig. 4B and Fig. 5B, right panels)?

This was only one experiment. It served to show that the variations of the transcription intensity in the control (the noise) were far exceeded by the variations in the experiment. 

Reviewer #3: 

(1) The Abstract remains almost unchanged. For example, it does not include mentioning the results of BrUTP experiments, which were included to the revised paper (lines 176-183). According to my opinion, these experiments are crucially important for interpretation of the FU-labeling data. The same is also true for the last paragraph of Introduction, where the authors summarize the main paper results.

We agree that these data are important, and we include them in the Abstract and Introduction of the second revised version.

. 

(2) Methods

It remains unclear, why the BrUTP labeling protocol is described only “briefly” (line 135). The publications cited by the authors for the scratch procedure used in the current study [55, 56] apply cell labeling with DNA replication but not with any transcription precursors.

Applying the scratch method to the labelling of transcription we used BrUTP instead of DNA precursors. Otherwise, the procedure was the same as in the replication labelling. We clarify this point in the revised text. 

What was the duration of cell fixation with methanol (line 133)?

The cells were fixed for 30 min. We provide this information in the revised text

What was the working concentration of BrUTP?

20µg/ml. We provide this information in the revised text. 

Line 140: “cy3-conjugated” should be replaced by “Cy3-conjugated”.

We make this correction in the revised text.

---

## [Editor Report · Decision Letter 2]

27 Jan 2020

Discontinuous transcription of ribosomal DNA in human cells

PONE-D-19-25095R2

Dear Dr. Smirnov,

We are pleased to inform you that your manuscript has been judged scientifically suitable for publication and will be formally accepted for publication once it complies with all outstanding technical requirements.

With kind regards,

Michal Hetman

Academic Editor

PLOS ONE
---

## [Editor Report · Acceptance letter]

20 Feb 2020

PONE-D-19-25095R2 

Discontinuous transcription of ribosomal DNA in human cells 

Dear Dr. Smirnov:

I am pleased to inform you that your manuscript has been deemed suitable for publication in PLOS ONE. Congratulations! Your manuscript is now with our production department. 

With kind regards,

on behalf of

Dr. Michal Hetman 

Academic Editor

PLOS ONE